# Gas Sensing Performance and Mechanism of CuO(*p*)-WO_3_(*n*) Composites to H_2_S Gas

**DOI:** 10.3390/nano10061162

**Published:** 2020-06-13

**Authors:** Fang Peng, Yan Sun, Weiwei Yu, Yue Lu, Jiaming Hao, Rui Cong, Jichao Shi, Meiying Ge, Ning Dai

**Affiliations:** 1State Key Laboratory of Infrared Physics, Shanghai Institute of Technical Physics, Chinese Academy of Sciences, Shanghai 200083, China; zpfzyyx@163.com (F.P.); taylorfish@163.com (W.Y.); Lyue8730@163.com (Y.L.); jiaming.hao@mail.sitp.ac.cn (J.H.); congrui@mail.sitp.ac.cn (R.C.); 2School of Electronic Electrical and Communication Engineering, University of Chinese Academy of Sciences, Beijing 100049, China; 3School of Materials Science and Engineering, Shanghai Institute of Technology, Shanghai 200235, China; jcshi@sit.edu.cn; 4National Engineering Research Center for Nanotechnology, No. 28 East Jiang Chuan Road, Shanghai 200241, China; meiyingge@163.com; 5Hangzhou Institute for Advanced Study, University of Chinese Academy of Sciences, Hangzhou 310024, China

**Keywords:** CuO/WO_3_, CuS formation, H_2_S, *n*-*p* type, molar ratio, weakening mode

## Abstract

In this work, the compositional optimization in copper oxide/tungsten trioxide (CuO/WO_3_) composites was systematically studied for hydrogen sulfide (H_2_S) sensing. The response of CuO/WO_3_ composites changes from *p*-type to *n*-type as the CuO content decreases. Furthermore, the *p*-type response weakens while the *n*-type response strengthens as the Cu/W molar ratio decreases from 1:0 to 1:10. The optimal Cu/W molar ratio is 1:10, at which the sensor presents the ultrahigh *n*-type response of 1.19 × 10^5^ to 20 ppm H_2_S gas at 40 °C. Once the temperature rises from 40 °C to 250 °C, the CuO/WO_3_ (1:1) sensor presents the *p*-*n* response transformation, and the CuO/WO_3_ (1:1.5) sensor changes from no response to *n*-type response, because the increased temperature facilitates the Cu-S bonds break and weakens the *p*-type CuO contribution to the total response, such that the CuS bond decomposition by a thermal effect was verified by a Raman analysis. In addition, with a decrease in CuO content, the CuO is transformed from partly to completely converting to CuS, causing the resistance of CuO to decrease from increasing and, hence, a weakening mode of *p*-CuO and *n*-WO_3_ to the total response turns to a synergistic mode to it.

## 1. Introduction

Gas sensors are urgently applied in various application fields such as military and aerospace, industrial production and safety, environmental monitoring, medical diagnosis [1,2,3], etc. Metal oxide semiconductors (MOSs) gas sensors are widely recognized because of their advantages such as convenience to carry, small size, low cost, simple operation, high sensitivity, fast response, and long life [4,5,6]. Among these MOSs, tungsten trioxide (WO_3_), a typical *n*-type semiconductor with a narrow band gap (2.4 eV–2.8 eV) [7], has achieved superior sensing performances with a wide range of toxic gases (e.g., NO_2_, H_2_S, O_3_, H_2_, NH_3_, and so on) [8,9,10,11,12,13,14]. However, single-component WO_3_ generally suffers from low sensitivity and requires elevated working temperatures, which obviously limits the commercial use in the gas monitoring field. In order to improve the performance of WO_3_-based gas sensors, the surface of WO_3_ needs to be functionalized with different materials [15,16]. Currently, cupric oxide (CuO) has attracted more attention as one of the *p*-type sensorial materials [17]. Nanostructured CuO can be prepared as a sensing material solely [18,19] or function with various *n*-type MOSs, such as SnO_2_ [20], ZnO [21], WO_3_ [22], etc., especially for H_2_S gas detection.

Hydrogen sulfide (H_2_S) gas is an extremely poisonous and flammable gas to humans and the environment which is often produced in coal mines, sewage pits, or natural gas industries [23]. It can cause eye irritation, fatigue, headaches, poor memory, dizziness, olfactory paralysis, and respiratory distress upon human exposure above 10 ppm H_2_S. It even causes paralysis and death when the concentration of H_2_S exceeds 120 ppm [24]. Consequently, highly sensitive, reliable, and rapid H_2_S sensors are in great demand. 

As a promising H_2_S sensing material, the combination of CuO and WO_3_ can accelerate the change in conductance and enhance the H_2_S gas sensitivity [25]. The extraordinarily improved H_2_S sensing performance of CuO/WO_3_ hybrids is normally attributed to two behaviors—that is, the CuO reacts with H_2_S to generate metallic CuS [22,26], and the CuS formation causes the destruction of the interfacial barrier built between the CuO and WO_3_. We named these behaviors CuS formation and barrier modulation [22]. Although the two strategies are known to increase the performance of gas sensors, there are few reports about compositional optimization in such sensing mechanisms. Meanwhile, the reaction of H_2_S gas with chemisorbed oxygen species—that is, the H_2_S oxidation mechanism—also works in the sensing process. Although much attention has been turned to sensing mechanism research based on CuO/WO_3_ sensors, there are still many problems in the mechanism cognition to be solved. For example, (1) Yu has reported that micro-CuO modification in WO_3_ can significantly amplify the H_2_S sensing response [25], but how does the CuO content affect the response performances of CuO/WO_3_ hybrids? (2) *p*-type CuO and *n*-type WO_3_ interact with the reducing gas H_2_S; they show the opposite response types, and the final response is dominated by the weakening mode of *n*-type and *p*-type responses. What is the molar ratio of CuO and WO_3_ that exhibits a maximum weakening mode? (3) Can the CuS formation behavior decrease the sensor’s resistance? CuS has a metallic character with a good conductivity and low resistance [27]. What is the molar ratio of CuO and WO_3_ when the weakening mode of *p*-type CuO and *n*-type WO_3_ changes to the synergistic mode in the total response? 

Herein, this work makes up for above deficiencies in the compositional optimization and mechanism analysis. The dependence of the H_2_S sensing characteristics of CuO/WO_3_ composites on the CuO content has been systematically investigated, and the corresponding working modes of the sensing mechanisms are also discussed in detail.

## 2. Experimental Scheme

### 2.1. Synthesis 

All the reagents were of analytical grade with no further purification. An amount of 3.5 g of sodium tungstate dihydrate (Na_2_WO_4_·2H_2_O, A.R. Aladdin) was added to 50 mL of deionized water and stirred continuously. After the 50 mL, 1 M sulfuric acid (H_2_SO_4_, A.R. Aladdin) was gradually dissolved in the mixture under stirring for 30 min until the color turned from transparent to golden yellow. Then, 15 mL of a precursor solution was slowly poured into a 25 mL Teflon-lined stainless autoclave for a hydrothermal reaction at 180 °C for 12 h. Finally, the naturally cooled WO_3_ powder was obtained by ultrasonic cleaning, centrifugation, and air drying. 

An amount of 0.232 g of WO_3_ nanocubes was added to 50 mL of ethanol (C_2_H_5_OH, A.R. Aladdin) while stirring for 60 min, then 10 mL copper acetate monohydrate (Cu (CH_3_COO)_2_·H_2_O, A.R. Aladdin) solution was dissolved in the above dispersion. The obtained mixtures were injected into a Teflon-lined stainless autoclave and maintained at 140 °C for 4 h. The final naturally cooled resultants were centrifuged, washed, and dried. The CuO/WO_3_ composites with different Cu/W molar ratios (1:0.5, 1:1, 1:1.5, 1:2, 1:5 and 1:10) were prepared by adjusting the concentration of the (CH_3_COO)_2_ Cu solution. For comparison, pure CuO and WO_3_ samples were also prepared. 

### 2.2. Characterization

The morphology of the CuO nanoparticles were characterized by scanning electron microscopy (SEM, FEI Sirion 200, FEI, The Netherlands) under an operational voltage of 10 kV and transmission electron microscopy (TEM) with selected-area electron diffraction (SAED) (JEOL2100F, JEOL, Tokyo, Japan). The phase and crystallinity of the nanoparticles was analyzed by Powder X-ray diffraction (XRD, D/max-2600PC, Rigaku Corporation, Tokyo, Japan) with Cu Kα radiation (λ = 1.5406 Å). The chemical composition was measured by an X-ray photoelectron spectrometer (XPS, ESCALAB 250Xi, Thermo Scientific, Waltham, MA, USA) using Al KR X-rays as the excitation source. The CuS formation and its oxidation process were verified by Raman spectra (LabRam HR800 Ev, HORIBA Jobin Yvon, Paris, France) employed at the excitation wavelength of 532 nm.

### 2.3. Preparation and Measurement

The schematic diagram of the sensor elements [28] is shown in Figure 1. A proper amount of sample powder was uniformly coated on a ceramic tube as a sensing film layer. A Ni-Cr alloy resistor was set through the tube as a heater, and the operating temperature was adjusted by the heating voltage. In this scenario, *V*_C_ is the test voltage with a certain value of 5 V, *V*_H_ is the heating voltage modulated on the Ni-Cr coil, *V*_m_ is the heating pulse voltage, *V*_out_ is the output voltage on *R*_L_, and *R*_L_ is a load resistor in series with the gas sensor. With the recorded *V*_out_ values of the load resistor, the equivalent resistances of the sensor can be calculated.

The sensitivity of the gas sensors was measured by a WS-30A system (Winsen Electronisc Technology Co., Ltd., Zhengzhou, China). All the measurements were carried out under the same condition of about 40% relative humidity. The response performances were similar to typical *n*-type or *p*-type semiconductor metal oxides—that is, the resistance decreased (increases) when the *n*-type (*p*-type) semiconductor sensors were exposed to the reducing gas [29,30], which are named the *n*-type response and *p*-type response, respectively. For distinction, the *n*-type response was recorded as the positive ratio of R_a_/R_g_—the sensor’s baseline resistance in air (R_a_) divided by that in target gas (R_g_)— while the *p*-type response was recorded as the negative ratio of (−R_g_/R_a_) [31]. The response time and recovery time, respectively, refer to the times for the sensor output to reach 90% of its saturation after the injection and release of the target gas.

## 3. Results and Discussions

### 3.1. Morphological and Structural Characteristics 

The morphological and dimensions of WO_3_ nanocubes were modified by different contents of CuO nanoparticles, which were investigated by SEM images in Figure 2a–h. CuO nanoparticles show a stone-like morphology with a diameter of approximately 20–50 nm in Figure 2a, while WO_3_ nanocubes have a cubic shape with glazed and flat surfaces, and the lengths of the cube structures were estimated to be around 80–150 nm in Figure 2h. Moreover, the CuO nanoparticles are grown on WO_3_ surfaces and form an irregular rough layer, which makes it efficient to offer more surface defects and adsorption sites for the WO_3_ nanocubes. Obviously, the irregular morphology of the CuO/WO_3_ composites becomes less visible as the CuO content decreases.

In order to further observe the surface morphology features of the CuO/WO_3_ composites, we selected a CuO/WO_3_ (1:1) sample to measure the TEM images and mapping patterns, which are displayed in Figure 2i–l. Figure 2i reveals that the polycrystalline CuO nanoparticles are modified on the WO_3_ surface; the lattice fringes with inter-planar spacings of 0.365 and 0.377 nm respectively correspond to the (200) plane and (020) plane of monoclinic WO_3_ [32,33]. The CuO nanoparticles are marked by a white oval in Figure 2i, but its well-defined lattice fringe separation is too weak to be observed. Figure 2j–l demonstrates the area mapping of the Cu, O, and W element distributions; it indicates that the three elements are evenly distributed on the surface of the CuO/WO_3_ (1:1) sample. The results indicate the successful modification of CuO nanoparticles on the WO_3_ surface after a hydrothermal process.

XRD patterns were employed to examine the phase and structure of the CuO/WO_3_ composites. In Figure 3, the diffraction peaks of the WO_3_, mainly located at 23.2°, 23.7°, 24.4°, 26.7°, 28.8°, 33.4°, 34.2°, 35.5°, 41.9°, 50.0°, and 56.0°, respectively, are assigned to (002), (020), (200), (120), (112), (022), (202), (122), (222), (140), and (420) crystalline planes, matching well with the monoclinic structure of WO_3_ (JCPDS card no. 43-1035). This indicates that the WO_3_ is well crystallized and coincident with the TEM results. After the CuO modification, the XRD spectra show the diffraction peaks of the CuO (11-1), (111), and (20-2) lattice planes, which are indexed with monoclinic CuO (JCPDS card no. 48-1548). For the CuO/WO_3_ composites with a Cu/W molar ratio of 1:0.5, the diffraction peaks of CuO are detected at 35.6°, 38.7°, and 48.9°, of which the peak at 48.9° is extremely weak. Compared with the diffraction peaks of WO_3_, those of CuO are wide and weak due to the small grain sizes. Meanwhile, the XRD peak of the CuO (11-1) crystalline plane is overwhelmed by that of the WO_3_ (122) lattice plane. As the Cu/W molar ratio decreases further to 1:1.5, the XRD peaks of CuO (11-1), (111) and (20-2) weaken enough to be unobserved. This phenomenon is similar to the results reported by Luo, who indicated that the CuO phase is not detected until the CuO loading in the CuO/Al_2_O_3_ composites is higher than 11.1% [34]. The XPS spectra of CuO/WO_3_ (1:1) were adopted to confirm the surface chemical composition, as illustrated in Figure A1. Figure A1a reveals that the binding energies at 934.2 eV and 954.2 eV, respectively, correspond to the Cu 2p3/2 and Cu 2p1/2 core levels for CuO, and the two satellite peaks at ∼944 eV and ∼962 eV relate to the presence of Cu^2+^ [35,36]. Figure A1b presents that the binding energy at 37.4 eV and 35.2 eV are respectively due to the W 4f_5/2_ and W 4f_7/2_ core levels, confirming the existence of the W^6+^ state in the WO_3_ [37,38]. Thereupon, the XRD and XPS results evidence the successful preparation of the CuO/WO_3_ composites.

### 3.2. Sensing Performances Analysis

According to the previous work [25,39], the single-component CuO and CuO/WO_3_ composite sensors all present an extremely long recovery process in H_2_S sensing at low temperature because of CuS formation, since the Cu-S bonding is too strong to break at low temperatures. Applying a short electric current pulse accelerates the oxidation of CuS to CuO, and hence the sensors’ recovery process. Therefore, all the sensors in this paper were applied a heating pulse and they present stable sensing properties. 

Furthermore, the dynamic sensing processes of the CuO/WO_3_ composites to 4 ppm H_2_S gas at different detection temperatures are indicated in Figure 4. As the load resistor is in series with the gas sensor, the equivalent resistances of the sensor can be calculated from the *V*_out_ on the load resistor—namely, a decrease (or increase) in the sensor’s resistance leads to increase (or decrease) in *V*_out_. The response type can be recorded via the change in *V*_out_ after the sensor contacts with the target gas. Obviously, the CuO contents in the CuO/WO_3_ composites affect the sensor’s response type. As the Cu/W molar ratios decrease, the response changes from the *p*-type to *n*-type. Moreover, the working temperature also affects the response types of CuO/WO_3_ (1:1 and 1:1.5) sensors; such a phenomenon is in agreement with the results reported by Zhou [40].

The dependences of the response phenomena on the operating temperature and gas concentration are presented as histograms in Figure 5. As shown, apart from the CuO/WO_3_ (1:1 and 1:1.5) sensors, the sensors have good response performances to different concentrations of H_2_S gas. The *p*-type response weakens when the Cu/W molar ratio decreases from 1:0 to 1:1, then the *n*-type response enhances as the ratio further decreases from 1:1.5 to 1:10. When multi-CuO modification WO_3_ are exposed to H_2_S gas, the H_2_S oxidation behavior and CuS formation behavior collectively amplify the CuO resistance [28]. Besides this, the H_2_S oxidation behavior and barrier modulation together decrease the WO_3_ resistance. The final response is determined by the opposite resistance variation from *p*-type CuO and *n*-type WO_3_—that is, the weakening mode of *n*-*p* type.

CuO/WO_3_ (1:0), (1:0.5), and (1:1) sensors all display *p*-type responses. Because the amplified resistance due to CuO is weakened by the reduced resistance derived from WO_3_, CuO/WO_3_ (1:0.5) exhibits a poorer *p*-type response in comparison to CuO. Moreover, such a weakening mode is much more significant with an increase in the CuO content; CuO/WO_3_ (1:1) demonstrates the lowest *p*-type response at the temperature of 40 °C–150 °C. However, at a higher temperature of 250 °C, CuO/WO_3_ (1:1) presents a small *n*-type response, indicating that the contribution of WO_3_ excels that of CuO. Because it will accelerate the Cu-S bonds breaking and the Cu-O bonds combining at high temperatures, the generated CuS was converted to CuO simultaneously. Ultimately, the CuS formation is negligible, which greatly reduces the CuO contribution to the total response. Furthermore, the CuO/WO_3_ (1:1.5) sensor has no response at temperatures of 40 °C–80 °C, since the sensor’s resistance is basically constant—that is, the weakening mode of *n*-*p* type maximizes at such temperatures. In addition, it has a very weak *n*-type response at temperatures of 150 °C–250 °C; this phenomenon is analogous to that of the CuO/WO_3_ (1:1) sensor at 250 °C.

The CuO/WO_3_ (1:2, 1:5, and 1:10) sensors all present the *n*-type response. As seen, the CuO/WO_3_ (1:10) sensor presents the highest gas response, and this suggests that micro-CuO modification greatly improves the sensing performance of WO_3_, which is consistent with the phenomena reported previously [22,25]. The sensor’s response reaches 1.19 × 10^5^ to 20 ppm H_2_S gas at 40 °C; at such low temperatures, it can achieve 798 even toward 0.2 ppm H_2_S gas. Our sensors have excellent sensing performances at low temperatures relative to the results of WO_3_-based nanosensors reported previously [16,41,42,43,44], as shown in Table A1 of the Appendix A. Figure A2 demonstrates the dynamic process of the CuO/WO_3_ (1:10) sensor under varying H_2_S gas concentration at 40 °C. As shown in Figure A2a, four response cycles are present consecutively at different concentrations of H_2_S gas. Therefore, the CuO/WO_3_ (1:10) sensor exhibits a good and competitive sensing performance at 40 °C. The response times reduce from 176 s to 2 s as the H_2_S concentration increases from 0.2 ppm to 20 ppm; after applying a heating pulse, the recovery times are improved significantly and basically within 30 s, indicating the rapid recovery feature in Figure A2b. The selectivity was studied by measuring the device response to acetone, ammonia, methanol, isopropyl alcohol, methylbenzene, xylene, and H_2_S in Figure A2c. The sensor sensitivity is about 4.5 × 10^4^ to 4 ppm H_2_S, but it is only around single digits towards 50 ppm of other gases, which indicates that the sensor is much more sensitive to H_2_S gas. The continual short-term tests were measured for 10 days upon 4 ppm of H_2_S gas in Figure A2d, showing that the sensor exhibits a good stability at 40 °C.

The markedly enhanced response of the CuO/WO_3_ (1:10) sensor generally owes to a significant reduction in the sensor’s resistance after micro-CuO is sulfurized to metallic CuS. Meanwhile, the CuS formation absolutely destroys the interfacial barrier constructed between the *p*-type CuO and the *n*-type WO_3_ [45,46]. This is the opposite of the case for multi-CuO/WO_3_ composites where the CuS formation increases the resistance of CuO, because multi-CuO is partly converted into metallic CuS and the carrier flows thorough the junction of the CuO/CuS structures, but micro-CuO is completely converted to CuS, facilitating the carrier to flow directly from the surface of the CuS layer. The phenomenon was discussed in our previous work [28]. For the CuO/WO_3_ (1:10) sensor, the *p*-type CuO and *n*-type WO_3_ synergistically amplifies the *n*-type response. Compared with WO_3_, the responses of CuO/WO_3_ (1:10) are much better, but those of CuO/WO_3_ (1:2) and CuO/WO_3_ (1:5) are poorer upon exposure to H_2_S gas, signifying that micro-CuO modification can promote but multi-CuO modification hinder the *n*-type response of WO_3_, since there is a weakening mode of multi-CuO and WO_3_ to the total response and a synergistic mode of micro-CuO and WO_3_ to it.

From response histograms in Figure 5, we can see that the response is enhanced with the increasing gas concentration for the CuO/WO_3_ (1:0, 1:1.5, 1:2 1:5, 1:10, and 0:1) sensors, because a lower coverage of gas molecules leads to lower gas response. A rise in the concentration of H_2_S increases the reactions benefiting from larger coverage, presenting an enhanced gas response approximately linearly with the gas concentration [47]. However, for the CuO/WO_3_ (1:0.5 and 1:1) sensors, the dynamic curves show a unique feature. As demonstrated in Figure 6 at 40 °C and 80 °C, the sensor’s response became weaker at the increase in H_2_S concentration, since the decline to *V*_out_ is smaller after exposure to the H_2_S gas. This phenomenon violates the linear relationship between the response value and the gas concentration. This may be due to the fact that CuO partly forms CuS upon low concentrations of H_2_S gas; the CuS formation mechanism and H_2_S oxidation mechanism together increase the sensor’s resistance, and hence the *V*_out_ drops obviously. However, the CuS formation mechanism enhances upon a high concentration of H_2_S, the metallic CuS layer forms on the CuO surface, inhibiting the increase in the sensor’s resistance, and then *V*_out_ drops with a smaller slope. This further verifies the above explanation for the different influences of CuS formation on the device resistance. As presented in Figure 6 under 150 °C and 250 °C, the *V*_out_ is found to vary linearly with the H_2_S concentration again because the enhanced oxidation capacity of CuS at high temperatures is not conducive to CuS formation. Besides this, the dynamic equilibrium of the surface molecular reaction is usually obtained at high temperatures [12,48]. Thus, the linear relationship between the gas response and gas concentration occurs again dominated by the oxidation mechanism at such temperatures.

Figure 7 displays the responses of the CuO/WO_3_-based sensors as a function of operating temperature toward 0.2, 1, 4, and 20 ppm of H_2_S gas. The optimum working temperature of CuO is lower than that of CuO/WO_3_ (1:0.5) because it facilitates the CuS formation at low temperatures due to sulfurization [26,49], and CuO is more dominated by the CuS formation mechanism than CuO/WO_3_ is (1:0.5). On account of the weakening mode of the *n*-*p* type, CuO/WO_3_ (1:1) has no response at low temperatures and its best operating temperature tends to be high. For the CuO/WO_3_ (1:1.5, 1:2, 1:5, and 1:10) sensors, their optimal operating temperatures reduce with a decrease in the CuO content; the reason for this may be that the interface barrier of the micro-CuO/WO_3_ composites is more easily broken upon H_2_S injection. The CuO and CuO/WO_3_ (1:1 and 1:5) sensors all display the two optimal working temperatures. The H_2_S oxidation dominates the sensing process at higher temperatures and the CuS formation plays a main role at lower temperatures, since the O-S exchange reaction is spontaneous and the oxygen ionosorption process is suppressed at low temperatures. On the other hand, the barrier destruction and weakening mode of the *n*-*p* type also work in the sensing process.

### 3.3. Sensing Mechanism

Raman spectroscopy is exploited in each step of the sensing process in order to confirm the CuS formation during the sensing process. As shown in Figure 8, the black lines are raw data measured under varied intensities of laser radiation. Curve (1) is the Raman spectrum of the pristine WO_3_ sample, which has been annealed at 500 °C in ambient condition for one hour, where the main bands located at 806 and 717 cm^−1^ correspond to the O-W-O stretching modes and the peaks at 326 cm^−1^ and 272 cm^−1^ are identified as O-W-O deformation modes [50,51]. All the Raman peaks of the WO_3_ sample are sharp and strong. The broad bands below 500 cm^−1^ could be deconvoluted into several peaks due to deformation and lattice vibrations. The main bands of curve (2) located at 292, 343, and 633 cm^−1^ evidence the presence of CuO [52]. When the CuO/WO_3_ (1:1) sample is exposed to H_2_S gas under 2 mW laser radiation, as shown by the curve (3), a very sharp band is observed at 472 cm^−1^, which is the signature of Cu-S bonding [53,54]. As stated previously, the high temperature facilitates the breaking of the Cu-S bonds, which weakens the CuS formation contribution to the *p*-type response of CuO. To understand the heating effect on CuS decomposition, the power of the laser radiation is increased from 2 to 14 mW to mimic the elevated operating temperature. The local temperature of the area under laser irradiation could rise. After “increasing temperature”, the intensity of the Raman peak at 472 cm^−1^ decreases gradually and finally vanishes, as shown by the curve (3)–(9). This evidences the breaking of Cu-S bonds due to the thermal effect of the laser beam. 

Furthermore, Figure 9 shows the baseline resistances of eight samples versus operation temperatures. As the temperature increases, the baseline resistances of all the samples decrease gradually; the phenomenon is consistent with the resistivity variation in the semiconductor materials on the temperature [55]. As mentioned above, the *n*-type (or *p*-type) response corresponds to a decrease (or an increase) in the sensor resistance after exposure to H_2_S gas. The temperature-dependent chemisorbed oxygen species (i.e., O_2_^−^, O^−^, O^2^^−^) [56] and the H_2_S molecules’ coverages are certain at the same working temperature and H_2_S concentration. For the *n*-type samples, their baseline resistances are larger, and their device resistances drop more significantly in general and hence their sensing responses are much higher. Therefore, the CuO/WO_3_ (1:10) sensor processes the strongest response signal, followed by the WO_3_ and CuO/WO_3_ (1:5) sensors, which have stronger response signals than the CuO/WO_3_ (1:2) sensor, and the CuO/WO_3_ (1:1.5) sensor has the weakest *n*-type response. Contrary to those of the *n*-type samples, the lower baseline resistances of the *p*-type samples upon exposure to H_2_S gas usually lead to a greater resistance variation and hence a more improved *p*-type response; thus, the CuO sample has the higher *p*-type response than the CuO/WO_3_ (1:0.5), and CuO/WO_3_ (1:1) has the lowest *p*-type response.

The sensing process in the CuO/WO_3_ composites with different CuO contents is schematically illustrated in Figure 10. It is assumed that the resistance of the macroscopic sensor is a combination of those of countless microstructures; the *p*-CuO/*n*-WO_3_ nanostructure, on exposure to air, could be looked upon as that the total resistor (R) results from the surface resistances (R_3_) connected in series with the effective resistance (the result of the parallel combination of the surface resistance (R_1_) and bulk resistance (R_2_). As exhibited in Figure 10a,b, a lot of oxygen molecules in air are chemisorbed on the surfaces of the CuO/WO_3_ composites based in depletion theory. These adsorbed oxygen molecules become ionized oxygen species O^δ^^−^ (e.g., O_2_^−^, O^−^, and O^2−^) by grasping free electrons from the conduction band of nanomaterials. When the *n*-type WO_3_ and the *p*-type CuO are brought into contact in the air, heterojunction barriers were constructed between the CuO nanoparticles and WO_3_ nanocubes by consuming a lot of electrons from WO_3_.

As exhibited in Figure 10c, when the composites with micro-CuO are exposed to H_2_S gas, the H_2_S molecules react with CuO to transform into metallic CuS completely, and the related reaction can be written as follows:(1)$$CuO(s)+H_{2}S(g)\to CuS(s)+H_{2}O(g)$$

The surface resistances R_1_ and R_3_ reduce sharply, since CuS owns a metallic character with a good conductivity and low resistance [20,27]. The potential barrier at the interface of the composites collapses due to the CuS formation; meanwhile, the chemisorbed O^δ–^ reacts with H_2_S to transform into H_2_O and SO_2_ according to the following reaction: (2)$$H_{2}S(gas)+3O^{\delta -}(ads)\to H_{2}O(gas)+SO_{2}(gas)+3\delta e^{-}$$

Those electrons previously trapped by the barrier construction and oxygen atoms are instantaneously released back into the conduction band of WO_3_, cause the bulk resistance R_2_ to decrease significantly. Ultimately, the sensor presents an excellent *n*-type response due to the R_gas_ being obviously lower than the R_air_. As displayed in Figure 10d, when the composites with multi-CuO reacts with H_2_S gas, CuO partly transforms to CuS with a lower work function [57], electrons would flow from the CuS to CuO and recombine with the holes in the valance band, resulting in an increase in electrical resistance of the CuO nanoparticles. Meanwhile, the H_2_S oxidation behavior releases the electrons from the surface states to the conduction band of CuO, leading to a further increase in the resistance of CuO [20]. Thereupon, the R_1_ and R_3_ are augmented simultaneously. However, both the barrier destruction and H_2_S oxidation process transfer plenty of electrons to WO_3_, causing a drastic reduction in the R_2_. As a result, the total resistance variation determined by the three resistances is uncertain; the sensor would present an *n*-type response when the R_gas_ is less than the R_air_, generally for WO_3_-rich composites. On the other hand, the sensor would indicate a *p*-type response when the R_gas_ is larger than the R_air_, usually for CuO-rich composites. 

## 4. Conclusions

In this research, CuO/WO_3_ nanostructures with different molar ratios were synthesized using a facile two-step acid hydrothermal method. Gas sensors based on the CuO/WO_3_ composites show different response phenomena to H_2_S gas based on the combined action of different working mechanisms. These are the CuS formation mechanism, the H_2_S oxidation mechanism, barrier modulation, and weakening mode or synergetic mode of *n*-*p* type. Accordingly, the sensing characteristic changes from the *p* type to *n* type as the Cu/W molar ratio decreases from 1:0 to 0:1. The weakening mode of the *n*-*p* type maximizes in the CuO/WO_3_ (1:1.5) sensor, and the working temperature changes the response type of the CuO/WO_3_ (1:1) sensor. Moreover, micro-CuO modification can promote but multi-CuO modification hinders the *n*-type response of WO_3_, among which the CuO/WO_3_ (1:10) sample presents an ultrahigh *n*-type sensing performance; the weakening mode of the *n*-*p* type changes to the synergistic mode of that. A gas-sensing mechanism analysis of the microstructure supplied abundant cognition for the different response phenomena of the macroscopic sensors. The Raman measurement verified that the formation of the CuS bonding and its decomposition can be effectively triggered by an elevated temperature.

## Figures and Tables

**Figure 1 nanomaterials-10-01162-f001:**
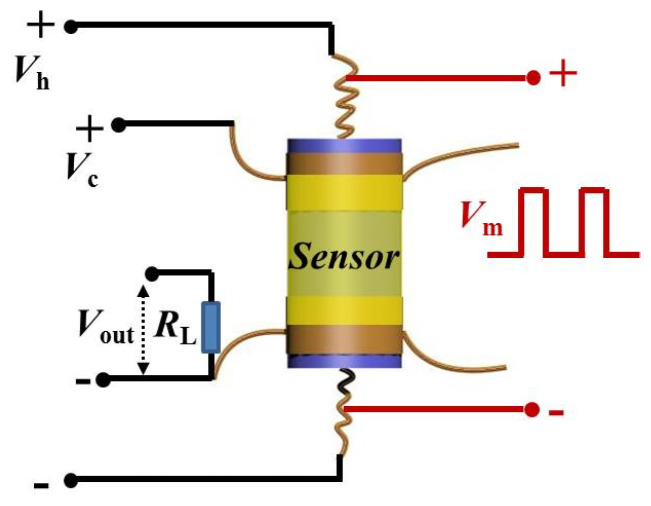
Schematic diagram of the gas sensor with electric modulation voltage.

**Figure 2 nanomaterials-10-01162-f002:**
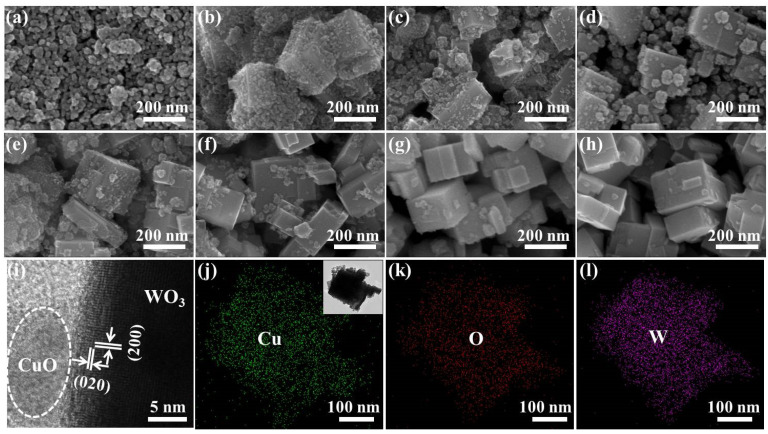
SEM images of the CuO/WO_3_ composites with different Cu/W molar ratios: (**a**) 1:0, (**b**) 1:0.5, (**c**) 1:1, (**d**) 1:1.5, (**e**) 1:2, (**f**) 1:5, (**g**) 1:10, and (**h**) 0:1. (**i**) TEM image of the CuO/WO_3_ (1:1) sample; (**j**–**l**) Cu, O, and W elemental mapping of the CuO/WO_3_ (1:1) sample.

**Figure 3 nanomaterials-10-01162-f003:**
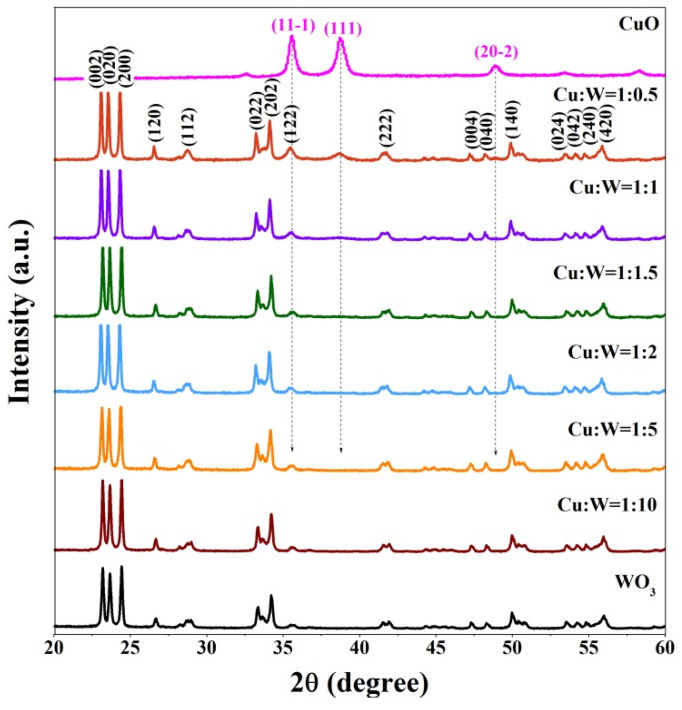
XRD patterns of the CuO/WO_3_ composites with different Cu/W molar ratios.

**Figure 4 nanomaterials-10-01162-f004:**
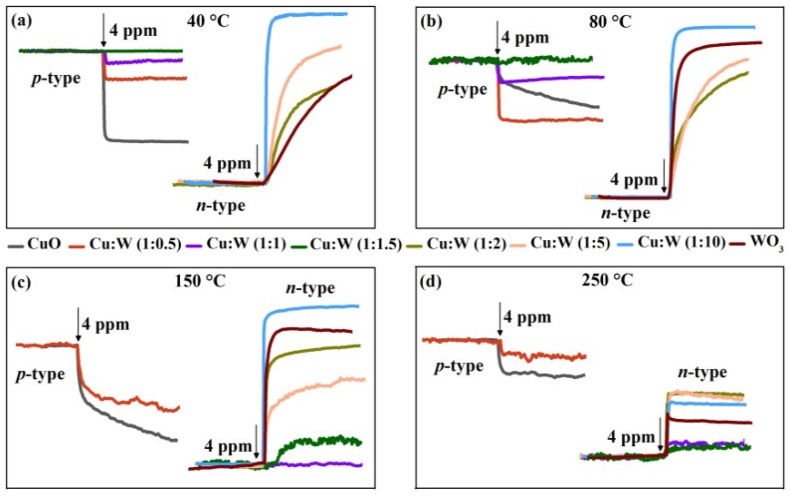
Dynamic curves of the CuO/WO_3_ composites with different Cu/W molar ratios upon 4 ppm H_2_S gas at (**a**) 40 °C, (**b**) 80 °C, (**c**) 150 °C, and (**d**) 250 °C.

**Figure 5 nanomaterials-10-01162-f005:**
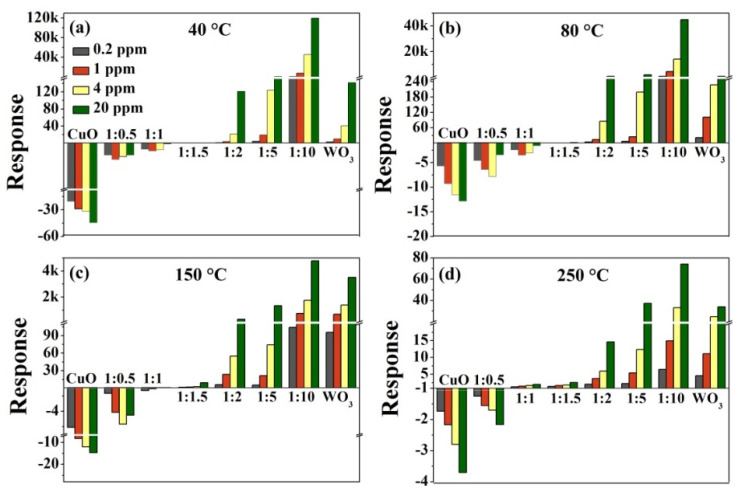
Responses of the CuO/WO_3_-based sensors toward 0.2, 1, 4, and 20 ppm of H_2_S gas at (**a**) 40 °C, (**b**) 80 °C, (**c**) 150 °C, and (**d**) 250 °C (a negative value implies the *p*-type response and a positive value represents the *n*-type response).

**Figure 6 nanomaterials-10-01162-f006:**
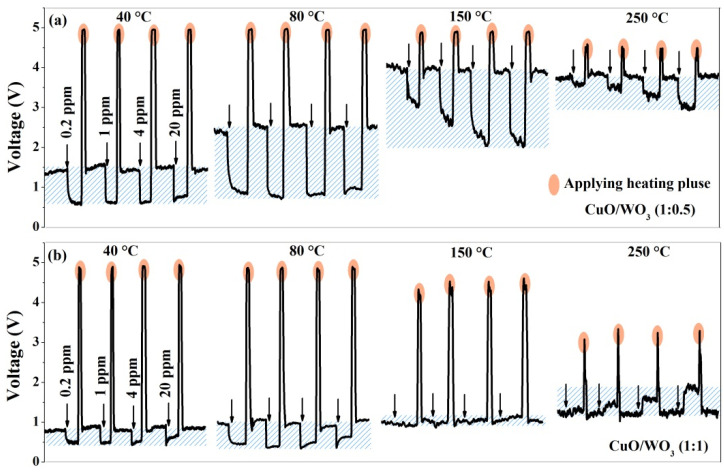
Dynamic curves of (**a**) the CuO/WO_3_ (1:0.5) sensor and (**b**) the CuO/WO_3_ (1:1) sensor to 0.2, 1, 4, and 20 ppm of H_2_S gas at different temperatures.

**Figure 7 nanomaterials-10-01162-f007:**
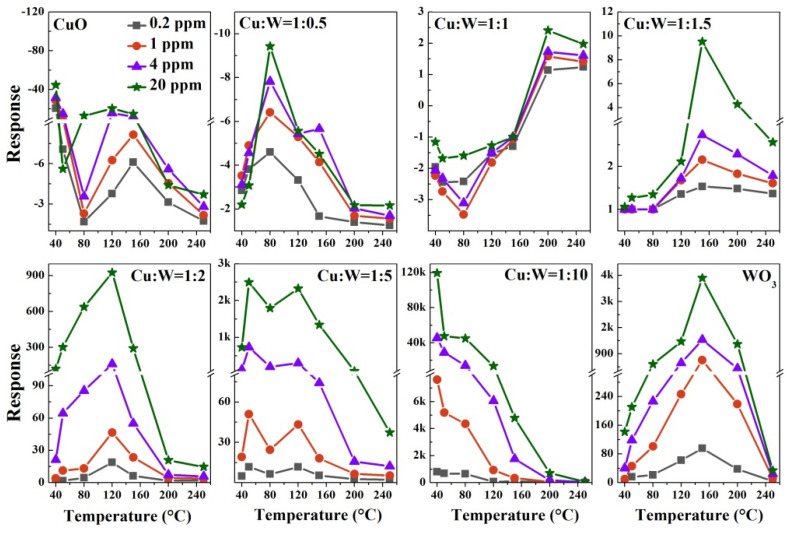
The responses of the CuO/WO_3_ composites to H_2_S gas as a function of the operating temperature.

**Figure 8 nanomaterials-10-01162-f008:**
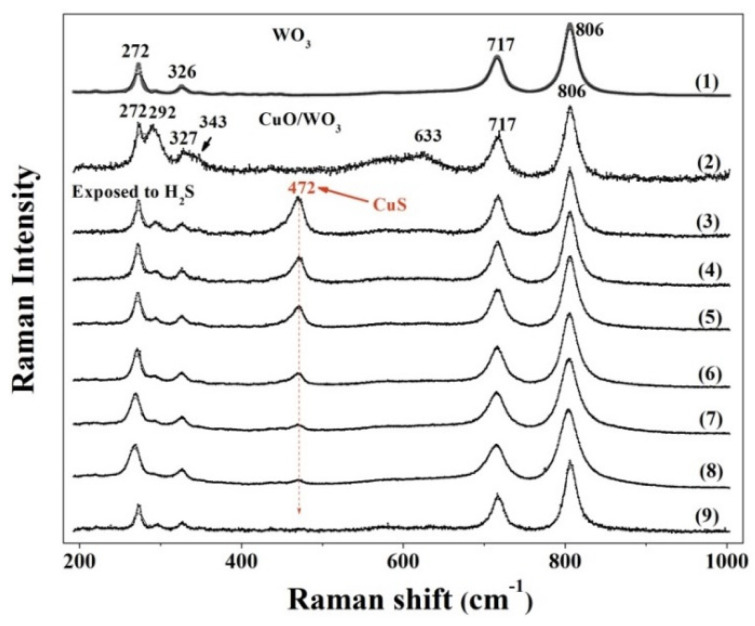
Raman spectra of the WO_3_ sensor (1) and CuO/WO_3_ (1:1) sensor (2) measured by 2 mW laser radiation exposed to air. The CuO/WO_3_ (1:1) sensor is exposed to H_2_S gas after being irradiated by the laser with different energies: (3) 2 mW, (4) 4 mW, (5) 6 mW, (6) 8 mW, (7) 10 mW, (8) 12 mW, and (9) 14 mW.

**Figure 9 nanomaterials-10-01162-f009:**
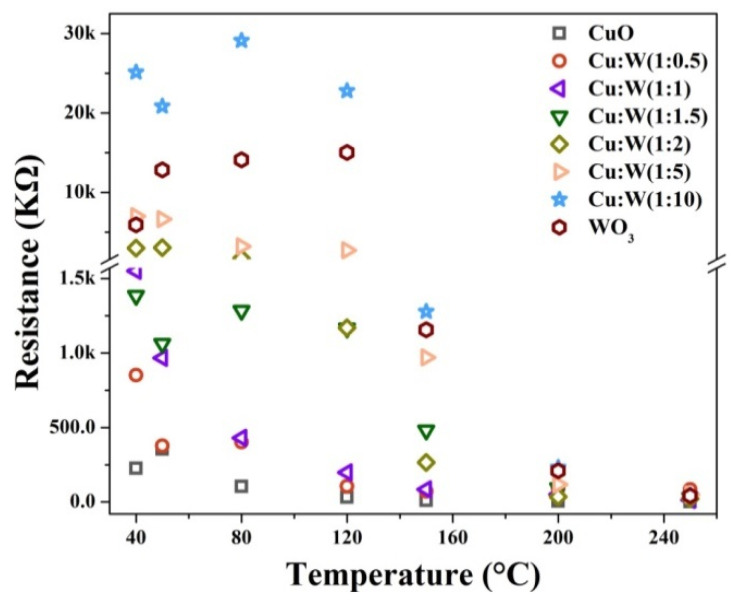
The baseline resistance values of the CuO/WO_3_ composites at different operating temperatures.

**Figure 10 nanomaterials-10-01162-f010:**
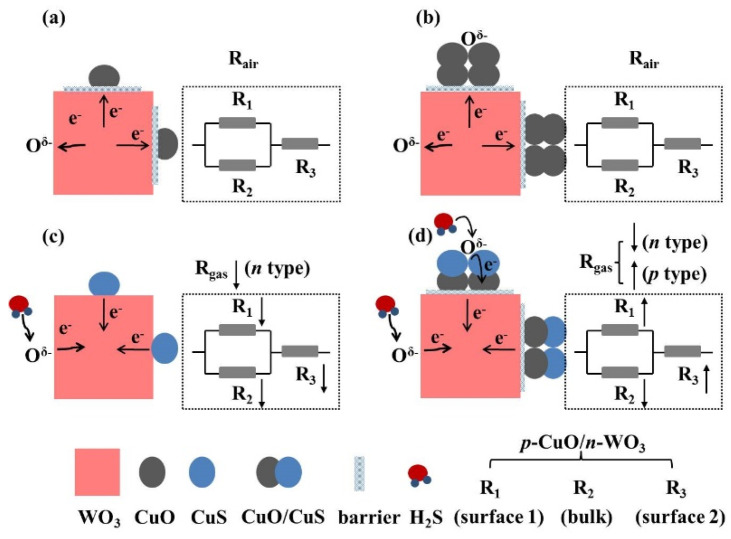
Schematic diagrams of the CuO/WO_3_ microstructure. In air: (**a**) low CuO content and (**b**) high CuO content. In H_2_S: (**c**) low CuO content and (**d**) high CuO content.

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
