# Peer review of "Gas Sensing Performance and Mechanism of CuO(p)-WO3(n) Composites to H2S Gas"

_nanomaterials, 2020, doi:10.3390/nano10061162_

Round 1

Reviewer 1 Report

In the manuscript nanomaterials-833152, the authors have prepared CuO/WO3 composites for hydrogen sulfide (H2S) sensing. The manuscript is interesting and I recommend a ‘minor revision’, however the authors need to answer the following concern:

  1. Why the authors have chosen H2S gas.? An explicit description for the need to detect H2S gas should be made in the Introduction section.
  2. “….working modes of sensing mechanisms are also detailedly discussed”. Check the word ‘detailedly’.
  3. Discuss the disappearance of some peaks in Figure 2.
  4. Discuss the schematic of sensor in section 2.3. It is hard to understand the substrate, and sensing chamber from the given statement.
  5. The H2S response in the presence of humidity should be provided to understand the effect of %RH.
  6. Important sensing parameters are missing from the work e.g. response/recovery time, long term stability and selectivity.
  7. A comparison table regarding the H2S sensing performance with similarly reported materials should be provided.
  8. The "Introduction" section should be improved. Some important references regarding the CuO and WO3 based sensors can be cited. e.g. https://aip.scitation.org/doi/10.1063/1.5123479; https://doi.org/10.1039/C8TA02702A; https://doi.org/10.1016/j.snb.2017.06.179

Author Response

Dear Editor,

Thank you for your email about our manuscript (nanomaterials-833152) for Nanomaterials. According to the comments and suggestions from you and referees, the paper is reorganized and the explained sentences and words are marked with highlight in the revised paper. The responses to reviewers' comments are shown as follows.

We are looking forward to hearing from you soon!

Sincerely yours

Yan Sun

Reviewer 2 Report

The paper entitled: "Gas sensing performance and mechanism of CuO(p)-WO3(n) composites to H2S gas" submitted by Peng et al. present the compositional optimization in copper oxide/tungsten trioxide (CuO/WO3) composites for hydrogen sulfide (H2S) sensing. The paper is well-written, well-organized and contents a sufficient data, however, some minor remarks prior to the publication:

Introduction:
Line 33 <gas> instead of <Gas>
if you mention that gas sensors are used in medical diagnosis (line 34) it is worth to provide the references for gases detected by such gas-sensitive layer, e.g. for WO3 (Performance of Si-Doped WO3 Thin Films for Acetone Sensing Prepared by Glancing Angle DC Magnetron Sputtering, IEEE Sensors Journal) as well as when CuO is mentioned as a good candidate for gas-sensing applications (line 43), the newest review could be presented (Coatings 2018, 8(12), 425; https://doi.org/10.3390/coatings8120425).
Line 64-67 should be rewritten as a new paragraph, for instance: "In this paper.... and the short summary of the paper would be beneficial for the readers".

Fig.1, the (j),(k), (l), do not provide any information, maybe the contrast could be increased, or please consider to use the negative option to change BLACK for WHITE.

Fig.2. <hkl> over the 50 deg. should be commented on as well.

Line 150-151-152, please provide a short explanation of why 4 ppm of H2S was used? as well as the information about the RH (Fig.3).

The gas-sensing protocol is missing. How the various H2S concentrations have been obtained (0,2/1/4/20 ppm)?

Apart from the above-mentioned remarks the paper is a really good paper, the experiments have been conducted correctly and the gas-sensing model, as well as schematic diagrams, showed in Fig.9 are well-done work.

Author Response

(The authors gave the same response as above.)
